# Associations between Self-Esteem, Psychological Stress, and the Risk of Exercise Dependence

**DOI:** 10.3390/ijerph18115577

**Published:** 2021-05-23

**Authors:** Frida Austmo Wågan, Monica Dahle Darvik, Arve Vorland Pedersen

**Affiliations:** Department of Neuromedicine and Movement Science, Norwegian University of Science and Technology (NTNU), N-7491 Trondheim, Norway; monica.darvik@ntnu.no (M.D.D.); arve.v.pedersen@ntnu.no (A.V.P.)

**Keywords:** exercise dependence, self-esteem, psychological stress, recreational training, exercise, self-worth

## Abstract

Body concerns and stress-related disorders are increasing in the younger population in a wide range of nations. Studies find links between both self-worth, exercise dependence, and self-esteem in relation to stress, but few have considered all three variables in relation to one another. The present study explored whether the co-appearance of high levels of psychological distress, and low levels of self-esteem may be a vulnerability factor for developing exercise dependence by studying the links between self-esteem, psychological stress, and exercise dependence. A standardized cross-sectional questionnaire was completed by 203 regular exercisers attending two gyms (mean age: 35.9 years). The variables self-esteem, psychological distress, and exercise dependence were all significantly correlated with each other, even after weekly exercise amount, age, and gender had been accounted for. Those who exercised for more than 9 h per week had a significantly higher score on stress and exercise dependence symptoms, and a lower score on self-esteem compared with the remaining groups. One could hypothesize that low self-esteem is a vulnerability factor and high psychological stress a maintenance factor for an exercise-dependent person. It is argued that more focus should be directed toward the negative consequences of excessive exercise.

## 1. Introduction

In times of uncertainty and stress, one will seek ways to master the situation [1,2], either in the form of adopting a changed focus and an escape from stressing thoughts and feelings, or through various coping strategies in terms of “behavioral responses that individuals use to manage or tolerate stress” [3], for example by taking control in other life-areas [4]. Poor self-esteem, whether general or specific, is associated with more maladaptive response patterns and poorer mental health when experiencing stress or difficulties [5,6]. For example, people with poor self-esteem or specific types of self-esteem are found to be more likely to seek coping strategies that distract them from what is perceived as difficult, and to avoid uncomfortable stress and negative emotions with a shift in focus [3,5] often referred to as “avoidance-coping” in the literature [7]. Such coping strategies may include excessive perfectionism in controlling external factors, status, and performance [8,9]; controlling food intake, weight, and body composition [10,11]; using alcohol intoxication for regulating emotions [2]; or controlling exercise and exercise performance, searching for the endorphins and good emotions that come from exercising [4,8], More adaptive or effective coping strategies in reducing emotional distress, and in turn leading to more positive mental health outcomes in the longer term are proposed as adherence to a healthy diet, spending time pursuing hobbies, or spending time outdoors [12].

Moderate amounts of exercise have been demonstrated to reduce both psychological distress [13,14,15] and depressive symptoms [16], to improve mental health [17], and to increase global and physical self-esteem [18,19,20,21]. Furthermore, physical activity (PA) and physical fitness are related to lower levels of perceived stress in the general population [22,23,24,25,26,27]. In a large population-based study including 1.2 million participants, [17] all forms of PA were found to be better for mental health than no PA at all, and the best effects on mental health were found in those who exercised 3–5 times a week for 30–45 min each session. Exercise levels below and above these parameters were linked to more days of poor mental health. Hence, the authors concluded that the dose–response curve has an inverted-U shape rather than a reversed L-shape, as outlined in the extant literature [28,29].

If the focus on exercise or the amount of exercise exceeds a certain level, the risk of physical overload and injury and/or psychological and social difficulties increase, as a result of great physiological and psychological demands on the individual, as well as conflicting time demands within other domains of daily life [1,4,30], Exactly how much exercise constitutes an excessive amount seems to vary between individuals, depending on the purpose of training [30,31,32,33], the individual’s age, their total stress in life [34,35,36], their gender [35,37], culture [38,39], exercise experience symptoms [40], and the context the activity is performed within [41,42]. Studies on larger populations have found that more than six hours’ weekly exercise increases the risk of exercise dependence [10,32], with a particular risk being associated with exercise levels above 10 to 12 h per week [8,36,43].

### 1.1. Exercise Dependence

To date, there are two common theories to try to understand exercise dependence: the interactional model [1] and the four-phase model for EA [4]. Both theories emphasize that excessive exercise may function as a strategy for coping with stress deriving from low self-worth and/or lack of mastery in other life areas. There is no standardized definition of exercise addiction [44] but there is a certain consensus that exercise addiction is a multidimensional phenomenon with psychological (e.g., experience of irritation and stress in the absence of exercise), behavioral (e.g., exercise frequency), and/or physiological dimensions (e.g., tolerance for increased amounts of exercise) [45]. Hausenblas and Downs’ [32] framework describes exercise addiction as a multidimensional and dysfunctional pattern of exercise, leading to clinically significant reductions in social, physical, and/or psychological functioning and/or persistent psychological stress as a result of the excessive focus on exercise. Their framework builds on seven diagnostic criteria for addictions, inspired by the DSM-IV criteria for substance-misuse disorders [46] proposed by Veale [47,48]. Reviews find an average prevalence of exercise addiction between 0.3% and 3.0% in normal populations and between 0.9% and 3.2% in active populations [48]. Also, a large variation is seen across studies due to different uses of measurements and conceptualizations of exercise dependence [49].

It is agreed that people who develop exercise addiction have a vulnerability factor, which may be personal or environmental [4], such as low levels of body-specific self-esteem [50] or perfectionism [8,9]. As with both substance abuse [2] and eating disorders [10], people with low self-esteem are at higher risk of developing unhealthy motives for and addiction to exercise [51]. People who largely build their identity around being an athlete or exerciser, exercising more, and spending more time on exercise are also more vulnerable to developing an addiction to exercise [31,52,53,54]. In particular, this seems to be the case for those who experience low global self-esteem [55].

### 1.2. Self-Esteem and Self-Worth

Global self-esteem can be seen as the total of positive and negative thoughts and emotions toward the self [5,56], including self-worth, which is about self-acceptance beyond achievements and looks [57], as well as self-evaluations; and specific self-esteem in smaller areas of life, like physical self-esteem, social self-esteem, appearance, and academic self-esteem [21,56,57,58]. This emphasis further underpins the notion that specific and global self-esteem are two related, but yet different, constructs, with a bidirectional effect on each other [5]. This in turn explains how physical activity may have a positive effect on global self-esteem by mastery and higher levels of physical self-esteem [21].

### 1.3. Psychological Distress

Models explaining exercise dependence point to psychological stress as a crucial factor in both the onset and the maintenance of an addiction to exercise [1,4,59]. Psychological distress is often described as the subjective feeling of stress inside the individual, occurring when he or she does not have adequate personal resources to meet situational demands effectively [3]. Previous research has found that individual variations in both the weekly PA level and self-esteem are related to fluctuations in psychological stress [14,60,61]. Overall, studies examining the effects of physical activity (PA) on both psychological and physiological responses to stress have found PA to lower the stress responses to some extent [62]. The fact that these conclusions are drawn from cross-sectional studies that do not control for all the confounding variables should be mentioned. More recent studies examining the link between PA and stress have found several confounding variables affecting the associations. For instance, Sibley, Hancock, and Bergman (2013) found motivation quality to be crucial to the effect PA had on students’ subjective experience of their own psychological distress. Østerås, Haga, and Sigmundsson (2017) found that mindfulness can modify the relationship between PA and reduction in stress levels in adolescents, while Lindwall and Lindgren (2005) found individuals’ subjective feeling of mastery in the activity as the most important factor in predicting the stress-reducing effects of PA [63,64].

Nevertheless, even when accounting for moderating variables like motivation and mastery, the risk of exercise dependence is independently associated with both psychological distress [59,65], and low physical [52,54] and global self-esteem [55]. However, to the authors’ knowledge, no studies have examined the relationship between psychological distress, self-esteem, exercise dependence, and exercise amount within the same group of individuals. Instead, they have tended to focus on associations between pairs of these constructs. Such an approach may obscure the shared variance between the constructs and mask a potential pathway between self-esteem and stress, resulting in a negative addiction to exercise. The current study investigated the hypothesis that high scores on exercise dependence would be associated with low scores on global self-esteem, high scores on psychological distress, and high weekly training volume—at the same time— in a sample of physically active individuals.

## 2. Materials and Methods

### 2.1. Participants

Two hundred and nine individuals were recruited from two different fitness centers. Six individuals (5 men and 1 woman) had incomplete answers and were therefore removed from further analysis (leaving *n* = 203). The lower age limit for participation was 16 years, and the average age of the sample was 35.9 years (SD = x; range: 16–71). Participants were selected on the basis of a desire to examine a wide and heterogeneous population, where age, life situation, exercise behavior, and motivation vary, in contrast to similar studies that have largely examined student populations [9,37,66,67,68]. Based on the existing literature on the association between training volume and risk of exercise dependence [8,10,32,36,43,69,70], the sample was categorized into three groups based on training volume. Group 1 consisted of those who exercised between 1 and 4.5 h per week (n = 93), group 2 individuals exercised between 5 and 8 h per week (*n* = 85), and those in group 3 exercised more than 9 h per week (*n* = 25).

Participants received information about the use of the questionnaires and GDPR and consented by completing the questionnaire.

### 2.2. Measures

A questionnaire consisting of the most widely used Norwegian version of the Rosenberg Self-Esteem Scale (RSES) [60,71,72]; a translated version of the Exercise-Dependence Scale (EDS), previously validated in Norwegian samples [10]; and a translated version of the Perceived Stress Scale (PSS) [73], validated in a Norwegian setting [74] was administered to potential participants by means of easy, accessible paper questionnaires, including information on the study, placed at the entrance/reception of the training center together with a safe box for completed questionnaires. No personal data whatsoever were collected in the questionnaire.

The RSES contains 10 items, while the EDS consists of 21 items categorized into seven subscales, each with three items: tolerance, withdrawal, intention effects, lack of control, reduction in other activities, time, and continuance. The PSS consists of 14 items. The internal consistence and reliability of the included measures was examined with Cronbach’s alpha. The RSES, the EDS, and the PSS proved to have adequate internal consistency of 0.94, 0.96, and 0.93, respectively.

The questionnaire also contained questions about exercise behavior; more specifically, these covered exercise volume (weekly exercise hours, rounded to the nearest half hour), exercise motivation, forms of exercise, and intensity of exercise. The questions on exercise volume and exercise behavior were based on standardized activity measures such as the Leisure-Time Exercise Questionnaire (LTEQ) [75], which has been frequently used in previous studies on exercise addiction [34,50,75], and is considered a sufficient target to distinguish between different activity levels in medium and large sample sizes [28,76].

### 2.3. Procedure

Participants were asked about global self-esteem (Rosenberg Self-Esteem Scale, RSES), exercise dependence (Exercise-Dependence Scale, EDS), stress (Perceived Stress Scale, PSS), and exercise behavior through self-reporting in a questionnaire. Participants were asked to mark responses on a scale of 1 (strongly agree) to 4 (strongly disagree) on the RSES, from 1 (never) to 6 (always) on the EDS, and from 1 (never) to 5 (very often) on the PSS. Each response was scored as 1, 2, 3, 4, or 5, and total scores were calculated by adding all values for each scale. Possible composite scores for RSES range from 10 to 40, with higher scores indicating higher global self-esteem [56,71] Five of the statements on the RSES are positively worded (statements 1, 2, 4, 6, 7), whereas the remaining five are formulated negatively (statements 3, 5, 8, 9, 10). Scores on positively worded statements are inverted when used for analysis. The same procedure was done with the PSS, as the PSS-scale consists of 14 questions in total, with seven of the questions on the scale positively worded and seven questions negatively worded. Possible composite scores for the EDS range from 21 to 126, with higher scores indicating higher levels of exercise addiction symptoms. Based on their total scores, participants are classified as asymptomatic (≤62), symptomatic (63–84), or at risk of becoming dependent on exercise (≥85). Possible total scores on the PSS range from 14 to 70, with higher scores indicating higher perceived stress [73]. Exercise behavior was recorded by reporting weekly exercise hours (full/half hours), exercise motivation (yes/no), exercise modes for which participants should circle one or two alternatives, and intensity (low/moderate/high). For some of the separate analyses, the sample was divided in three groups based on their weekly exercise amount, respectively between 1–4.5 h a week, between 5–8.5 h a week and above 9 h a week. This ranges were carefully selected based on an earlier review of the studies examining “threshold” training amounts in people with/and without exercise addiction [10,32,67,69,70].

The data collection took place from February to April 2019. Participants received information about the project, and about the use of the data, and consented by completing the questionnaire and posting it in a box behind the front desk of the training centers. No personal data whatsoever were collected. This was done in an attempt to ensure that participants would give more honest answers to the questions, and also as a means of increasing the number of participants in the study.

### 2.4. Data Analysis and Processing

IBM SPSS version 25.0 (IBM, Armonk, NY, USA) was used for all statistical analyses. Hausenblas and Downs’ (2002b) scoring manual was used.

## 3. Results

### 3.1. Demographics

A total of 203 participants (55% women and 45% men; mean age 35.9 years old) completed the questionnaire. Information about prevalence of ED (at risk, symptomatic, asymptomatic), weekly exercise amounts and mean scores on the outcome measures are presented in Table 1. Most participants (59.5%, *n* = 121) reported themselves as exercising mainly at moderate intensities, 34% (*n* = 69) reported themselves to be exercising at high intensity, and the remaining 6.5% (*n* = 13) exercised at low intensities. Resistance training was the most frequent form of exercise (46%, *n* = 92), followed by aerobic exercise performed as either running, biking, cross-country skiing, or cross-training (37.5%, *n* = 75). Only 16 participants (7.9%) exercised with the aim of performing well in a sport, while the remaining 187 participants (92.1%) reported that they exercised for fitness- and health-related reasons.

Based on the EDS criteria [32] 20 participants (9.9%) were classified as at risk, 38 participants (18.7%) as symptomatic, and the remaining 145 participants (71.2%) as asymptomatic. No gender differences were apparent in the numbers of participants classified as at risk of developing exercise dependence.

### 3.2. Relationships between Self-Esteem, Weekly Exercise, Stress, and Exercise Dependence

Spearman’s Rho correlations were computed between the variables self-esteem, weekly exercise, stress, and exercise-dependence symptoms (see Table 2). All outcome measures correlated significantly with each other (*p* ≤ 0.001). There was a negative correlation between self-esteem and EDS score, as seen in Figure 1, whereas a positive association was found between stress and EDS score, as seen in Figure 2, indicating that individuals with the highest EDS scores experienced more stress and lower levels of self-esteem than those with lower scores on the EDS. The EDS subcategories “withdrawal effect”, and “reduction in other activities” (items 5, 11, 19) had the strongest correlations (r ≤ 0.53) with overall scores on both self-esteem (RSES) and stress (PSS).

### 3.3. Group Differences in Exercise-Dependence Symptoms, Stress, and Self-Esteem

Significant differences between groups were observed for total scores on self-esteem, psychological distress, and exercise-dependence symptoms when we split the data by three different levels of weekly exercise times into 1–4.5 h, 5–8.5 h, and over 9 h per week. (see Table 3). Further analysis revealed no significant variance between participants in the 1–4.5 h and the 5–8.5 h per week groups. When the outcome scores for participants exercising more than 9 h per week were compared to the 1–4.5 h per week and the 5–8.5 h per week groups separately, the >9 h per week groups had significantly lower scores on self-esteem and higher scores on exercise-dependence symptoms and psychological distress than the remaining groups in all analyses.

### 3.4. Age and Gender Differences for Exercise-Dependence Symptoms, Stress, and Self-Esteem

Spearman’s Rho correlations were computed between age, self-esteem, weekly exercise, stress, and exercise-dependence symptoms. A Kruskal–Wallis test between the three different exercise volumes split by gender was performed to test whether the mean scores on self-esteem, psychological distress, and exercise dependence differed between genders with varying amounts of weekly exercise. Age correlated significantly (*p* ≤ 0.001) with total exercise amount (r = 0.28) and total score on the EDS (r = 0.50), but no correlations were seen between age and psychological distress or self-esteem. After adjusting the significance level on the Kruskal–Wallis test with Bonferroni correction, only the scores on psychological distress (z = −0.28) and weekly exercise volume (z = −0.259) differed between genders within the three exercise groups. Looking at the total sample (i.e., not split by weekly exercise volume), only psychological distress differed significantly between genders, but the difference was small to moderate (see Table 1).

## 4. Discussion

The aim of our study was to examine if the combination of low levels of self-esteem and high levels of psychological distress was related to high scores on the measures of exercise-dependence symptoms, to further predict risk factors for developing an addiction to exercise. In line with our main hypothesis, and based on the theoretical frameworks for understanding Exercise Dependence by Freimuth et al. (2011), and Szabo and Egorov (2013) [1,4], we found that higher levels of psychological distress and low levels of self-esteem were related to higher scores on measures of exercise dependence, as well as higher amounts of weekly exercise hours. More specifically, self-esteem had a significant negative correlation with both the Exercise-Dependence score and total weekly exercise volume, while psychological distress had a significant positive correlation with the Exercise-Dependence score and the total weekly exercise duration. All correlations were significant after accounting for age and gender, meaning there is a universal trend in both genders, in that those who had weekly exercise volumes above 9 h had significantly higher scores on exercise dependence and psychological distress, as well as lower scores on self-esteem compared to individuals exercising fewer than 9 h each week. Women generally had significantly higher scores on stress and a lower volume of weekly exercise. However, the differences were small to moderate. No gender differences were seen in either self-esteem or exercise-dependence symptoms.

The present findings expand previous research by showing that low self-esteem [52,54] and high levels of psychological distress [59,65] are co-existent in individuals with a high risk for developing exercise dependence. Such findings are also in line with qualitative studies on eating disorders [10,11] and exercise dependence [30,77] reporting excessive control over food or exercise as a coping strategy for handling psychological distress and a sense of inferiority compared to others. More specifically, the existing literature has reported that negative thoughts about one’s physical appearance is a source of negative stress, especially in young adults [78,79,80,81]. Excessive exercise as a means of attaining the perceived standards/ideals of what a body should look like may in part explain the reported correlations between low self-esteem, high weekly exercise-volume, and high scores on exercise-dependence symptoms. On the other hand, this cannot explain why global measures of self-esteem that measure general feelings of self-worth—and not bodily self-esteem or physical appearance—also seem to correlate with psychological distress and exercise dependence in the present sample of individuals, as well as in previous studies [33,80,82].

We found that both exercise-dependence symptoms and weekly exercise volume declined with age, as in earlier studies [34]. However, even after adjusting for both gender and age, the correlation between self-esteem, risk for exercise dependence, and psychological stress remained significant in all analyses. This stands somewhat in contrast to previous research that has reported both gender and age to influence scores on measures for self-esteem [79] and exercise dependence [34], prompting the authors to identify both gender and age as crucial factors for the onset of eating disorders [30], exercise dependence [34], or a combination of the two [10,11]. The discrepancy between the present and previous findings may be due to different ages of the studied populations across studies [34,79] as well as cultural, contextual, or individual variations [39,40]. The participants in our study exercised more than what other studies argue to be “an optimal volume of exercise” for the “normal population”, with a weekly exercise time of 5.4 h [17], 22–40 METS [83], or in a range from 2.5 to 7.5 h [84]. However, when compared with other studies examining exercise dependence [38,43,69,70], the mean amount of weekly exercise hours in the present sample is similar to previous findings. Many researchers have attempted to establish a “limit” for when exercise gets “unhealthy” and “addictive” [32,68], despite the rather similar mean exercise volumes per week in the existing literature, there is still no consensus about such a limit. Considering limits proposed in previous studies, such as 5–6 h/week [32,69,70] and 9–10 h/week [8,36,38,43], we chose to split the present sample into three different exercise groups (1–4.5 h per week, 5–8.5 h per week, and more than 9 h per week). In line with earlier results, we found that the group that exercised for more than 9 h/week differed significantly from the two remaining groups, in terms of exercise-dependence symptoms, self-esteem, and psychological distress parameters. In line with Chekroud’s et al. (2018) findings from examining 1.1 million individuals, our study supports the hypothesis that the relationship between exercise volume and health benefits looks more like an inverted U, with an optimal point before the effects slowly diminishes as the exercise volume increases [17]. Thus, our results do not support the assumption that has dominated the literature until rather recently [28,29] that the relation between the amount of exercise, and the physical- and mental-health outcomes form can be depicted as an inverted-L curve.

The present study found a prevalence of individuals at risk for exercise dependence (9.9%) which is comparable to previous research conducted in sports and exercise settings [32,70,85,86], while being higher than in studies conducted on larger “normal” populations, for example, representative populations for one country [44,87]. Furthermore, cultural and national differences have also been reported to play a role in terms of prevalence of exercise dependence [85] as well as self-esteem and the cultural emphasis on appearance and body shape [88,89], which must be taken into consideration when comparing our study to others.

Although our results to some degree expand the extant literature, limitations exist. Firstly, self-reported data on weekly exercise volume may have intrinsic limitations, such as social desirability [90]. Secondly, although this is one of the relatively few studies to examine the impact of both self-esteem and psychological distress on exercise dependence, the present results of this study are based on cross-sectional data, not observations of changes in stress and self-esteem due to changes in exercise volume, or exercise-dependence symptoms. Use of longitudinal data tends to provide better understanding, allowing attributions regarding the cause or directional effects. A major limitation of our study was that exercise motivation included only these two categories: desire to improve performance in sports/races and health- and fitness-related reasons, based on the findings of earlier studies, showing that exercise for health reasons is found to be related to improvements in body image [18,89,91] and self-esteem [18,64] during exercise interventions or regular physical activity. Other studies find that motives such as exercising to lose weight, to improve one’s appearance, or to avoid the feeling of inferiority compared to others are related to reductions in self-esteem [10,33,92]. Hence, we should have included a third category addressing appearance-based exercise motivation to separate those exercising for health, and those exercising for appearance. The present results therefore need to be seen in light of studies reporting activity motivation [83,92,93] and context [41,42] as crucial for determining the psychological effects of that specific activity on self-esteem. Yet further research is needed, addressing the role exercise motivation plays for the relationship between exercise dependence, psychological distress, and self-esteem.

Another aspect to take into consideration is that the EDS is a screening device developed to distinguish among at-risk, nondependent symptomatic, and nondependent asymptomatic individuals [94], and not a diagnostic system. Even if variables correlated with increased risk of developing exercise dependence may, in part, explain why some develop and maintain an addiction to exercise, clinical interviews by qualified personnel are needed to understand the mechanisms behind exercise dependence for each individual [87]. The present results enable no such diagnosing, nor do they in any other way suggest more severe mental illness. Furthermore, our research was limited by the methodology of the present study itself, as cross-sectional questionnaires are seen as inappropriate when talking about causal relationships between phenomena and related underlying mechanisms. Finally, research is needed on more of the mediating mechanisms underlying changes in self-esteem, psychological distress, and exercise behavior across the subject’s lifespan, such as the context of the activity, or the societal impact.

## 5. Conclusions

In summary, our study makes several significant contributions to the exercise-dependence literature. First by expanding the excitant literature by showing a co-existence of low self-esteem and high levels of psychological distress in individuals with a high risk for developing exercise dependence. Second by showing a further co-existence between both low self-esteem, high levels of psychological distress and high risk for developing exercise dependence in individuals with weekly training amounts beyond 9 h a week, even after adjusting for gender and age. Third, by supporting recent studies demonstrating an inverted U-relationship between weekly training amounts and mental health. Due to the limitations in the study, it is a first step in understanding variations in the varying effect of exercise on both stress and self-esteem.

## Figures and Tables

**Figure 1 ijerph-18-05577-f001:**
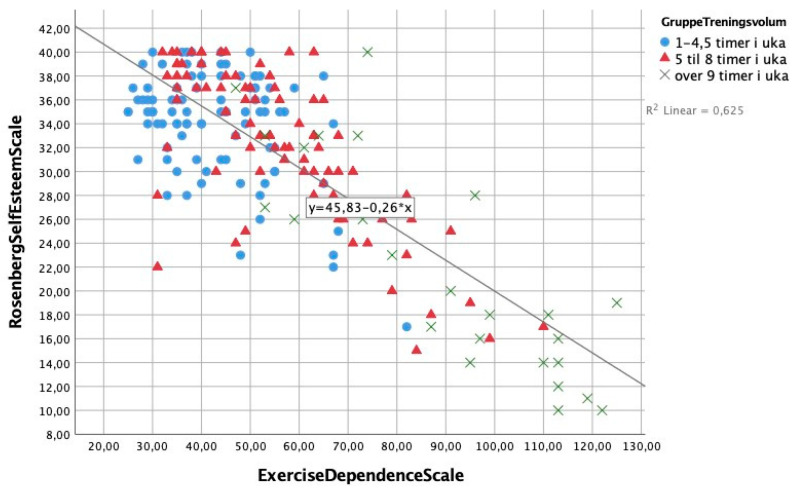
Graphic representation of the association between self-esteem (RSES-score) and exercise-dependence symptoms, by three different levels of weekly exercise hours.

**Figure 2 ijerph-18-05577-f002:**
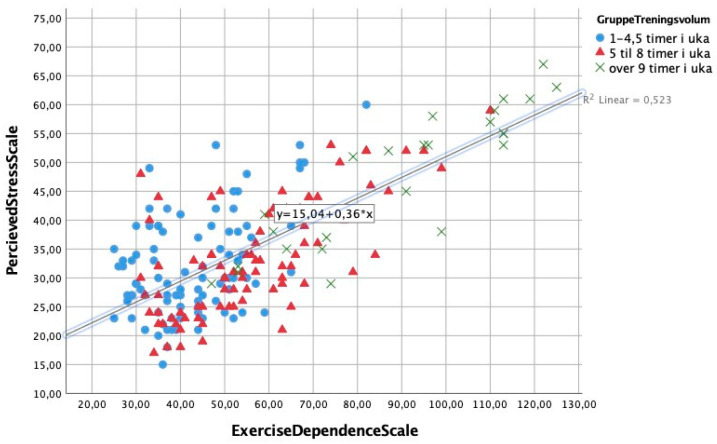
Graphic representation of the association between stress (PSS-score) and exercise-dependence symptoms, by three different levels of weekly exercise hours.

**Table 1 ijerph-18-05577-t001:** Descriptives and outcome measures by gender.

Variable	Total (*n* = 203)	Men	Women	Range	Z
Exercise (Hours/Week)	5.47 (SD = 3.0)	5.9 (SD = 2.9)	5.1 (SD = 3.0)	1–21	−2.59 **
Self-Esteem (RSES)	31.8 (SD = 7.1)	32.7 (SD = 7.25)	31.1 (SD = 6.9)	10–40	−1.53
Stress (PSS)	34.5 (SD = 10.8)	32.1 (SD = 9.8)	36.5 (SD = 11.2)	14–70	−2.85 **
Exercise Dependence (EDS)	54.4 (SD = 21.8)	54.5 (SD = 20.9)	54.0 (SD = 22.4)	21–126	−0.33

** Significant *p* ≤ 0.005.

**Table 2 ijerph-18-05577-t002:** Correlation analyses.

	Self-Esteem (RSES)	Exercise Dependence (EDS)	Weekly Training Hours	Stress (PSS)
(RSES)	1			
(EDS)	−0.597 **	1		
Weekly Training Hours	−0.338 **	0.566 **	1	
(PSS)	−0.781 **	0.567 **	0.298 **	1

** Significant correlation *p* ≤ 0.001.

**Table 3 ijerph-18-05577-t003:** Between-group differences in self-esteem, stress, and exercise-dependence scores.

	1–4.5 h/Week (*n* = 93)	5–8.5 h/Week (*n* = 85)	>9 h/Week (*n* = 25)
RSES (Mean Score, SD)	34.2 (SD = 4.4)	32.2 (SD = 6.4)	21.9 (SD = 9.0)
EDS (Mean Score, SD)	42.9 (SD = 11.7)	56.4 (SD = 17.2)	89.6 (SD = 24.4)
PSS (Mean Score, SD)	31.9 (SD = 8.9)	33.6 (SD = 9.7)	47.5 (SD = 11.9)

## Data Availability

The data presented in this study are available upon reasonable request from the corresponding author.

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
