# Peer review of "Associations between Self-Esteem, Psychological Stress, and the Risk of Exercise Dependence"

_ijerph, 2021, doi:10.3390/ijerph18115577_

Round 1
Reviewer 1 Report
Thanks for giving me the chance to review the manuscript ‘Associations between self-esteem, psychological stress, and the 2 risk of exercise dependence’. I think it is interesting. The topic is timely, and method and result sections are sound.
The idea of the manuscript was to proof if there is a significant correlation between self-esteem, psychological stress, and risk of exercise dependence. They filled a gap for self-esteem, psychological stress, and risk of exercise dependence. The major strength is whether low self-esteem may be a vulnerability factor for developing exercise dependence by studying the links between self-esteem, psychological stress, and exercise dependence. Thanks to the manuscript, the connection between self-esteem, psychological stress, and exercise dependence could be built. It should be noted that the studies are well done and the process of the studies is reasonable. Nevertheless, I have some further minor amendments.
1.The review of the literature in this manuscript is sufficient,but does not make a comprehensive and in-depth analysis of the theory frame in these issues, and it does not put forward the possible internal mechanism. So, the inherent logic and the hypotheses are lack of strong theory frame support.
- The core concepts of this study would be defined more clearly in this part.
- It is suggested that the relevant theories should be added to the discussion, and the advanced nature and creativity of the research should be explained from the theoretical support perspective.
Author Response
Response to reviewer 1:
- The review of the literature in this manuscript is sufficient,but does not make a comprehensive and in-depth analysis of the theory frame in these issues, and it does not put forward the possible internal mechanism. So, the inherent logic and the hypotheses are lack of strong theory frame support. The core concepts of this study would be defined more clearly in this part.
Response: To expand the theoretical framework underpinning the study, a few sentences describing the most used theories on the exercise-dependence field were added to the text (line 71-73) "Both theories emphasizing that excessive exercise may function as a strategy for coping with stress deriving from low self-worth and/or mastery in other life areas"
It is suggested that the relevant theories should be added to the discussion, and the advanced nature and creativity of the research should be explained from the theoretical support perspective.
Response: To further explain our findings in relation to excisting theories, the following was added to the text in the discussion-part (296-300): In line with our main hypothesis, and based on the theoretical frameworks for understanding Exercise dependence by Freimuth et al., (2011) and Szabo & Egorov (2013), we found that higher levels of psychological distress and low levels of self-esteem were related to higher scores on measures of exercise dependence, as well as higher amounts of weekly exercise hours.
Reviewer 2 Report
The paper presents an exploratory analysis between self-esteem, stress, and excercise dependence, and excercise duration. The analysis is conducted in a population of 203 individuals with a wide range of characteristics. Results are presented clearly, and are discussed adequately.
My main concern is the following, in Section 2.1: A total of 203 individuals participate in this study. They are split in 3 groups based on the duration that they excercise for every week. The three subgroups contain 93, 85, and 25 participants respectively. These sizes seem somewhat small, given the multiple parameters that these samples have regarding age, sex, life situation, excercise motivation, etc
Additionally, how have the three ranges (1-4.5, 5-8.5, >9) been selected? It would be interesting to see the distribution of excercise duration in a figure.
Finally, Figure 1: x-axis label is missing. Also the caption is probably wrong?
Author Response
Response to reviewer 2:
- My main concern is the following, in Section 2.1: A total of 203 individuals participate in this study. They are split in 3 groups based on the duration that they excercise for every week. The three subgroups contain 93, 85, and 25 participants respectively. These sizes seem somewhat small, given the multiple parameters that these samples have regarding age, sex, life situation, excercise motivation, etc.Additionally, how have the three ranges (1-4.5, 5-8.5, >9) been selected? It would be interesting to see the distribution of excercise duration in a figure.
Response: When the sample was split in to the three subgroups based on their weekly exercise-amount, we din not separate the subgroups any further by splitting in to categories as gender, age, exercise motivation etc. Further, the small and non-parametric sample was taken in consideration when choosing non-parametric tests in the analyses.
To explain why these three ranges was selected, the following was added to the text (218-221): "For some of the separate analyses, the sample was divided in three groups based on their weekly exercise amount, respectively between 1-4.5h a week, between 5-8.5h a week and above >9h a week. This ranges were carefully selected based on a earlier review of the studies examining “threshold” training-amounts in people with/and without exercise addiction (Kovacsik et al., 2018; Hausenblas & Downs, 2002a; Terry et al., 2004; Downs et al., 2004; Bratland-Sanda et al., 2011; Davis, 1990; Hagan et al., 2003; Lichtenstein et al., 2014; Meulemans et al., 2015).
- Finally, Figure 1: x-axis label is missing. Also the caption is probably wrong?
Response: The caption for fig. 1 was changed to "Graphic representation of the association between Self-esteem (RSES-score) and exercise dependence symptoms, by three different levels of weekly exercise hours.". X-axis label was added: "ExerciseDependenceScale".
Reviewer 3 Report
I would recommend having some changes.
Thanks

Author Response
Response to reviewer 3:
Abstract
1. Authors should include a little background of the study.
Response: A few sentences about the background/context for the study were added in line 9-10. "Body concerns and stress-related disorders are increasing in the younger population in a wide range of nations.”
2. [Line 17] It is not clear all main variables. I recommend the structure: The variables are….
Response: To make it more clear which main variables who are significantly correlated, we made a few changes in the text (line 17-18 and line 12-14). “The present study explored whether the co-apparance of high levels of psychological distress, and low levels of self-esteem may be a vulnerability factor for developing exercise dependence by studying the links between self-esteem, psychological stress, and exercise dependence.”…. "The variables self-esteem psychological distress and exercise dependence were all significantly correlated with each other, even after weekly exercise amount, age and gender had been accounted for".
3. Authors must specify the type of study design. A cross-sectional study was carried out with a sample questionnaire but how? Authors should speak about the sample a little more.
Response: Some details about the participants and the method was added to the text (line 16-17). More details are provided in the "Material and methods"-section because of limited space in the abstract.
Introduction
1. [Line 28] … the present time could be also COVID from that situation (Have you thought about it?) Good way to add some new references about it to enrichment your introduction.
Response: Thanks for your great advice for enriching our study! Unfortunately the data was gathered at time before the COVID-pandemic took place, and the pandemic is therefore not relevant to mention in this study.
2. [Line 30] you mention through coping and control in other areas. You could add the references about coping strategies for instance.. (suggestions in the PDF).
Response: A better definition of coping and a reference for this was added to the text (line 29-32). “either in the form of adopting a changed focus and an escape from stressing thoughts and feelings, or through various coping strategies in terms of “behavioral responses that individuals use to manage or tolerate stress” (cf. Lazarus & Folkman, 1984), for example by taking control in other life-areas (Freimuth et al., 2011)”
3. [Line 34] It is very interesting, and I will add some theory or at least mention which are the coping strategies such as, for example: there are some coping strategies that can be organized: problem-focused categories: strategies, which are directed at managing or emotion-focused altering which have strategies for regulating emotional and eating disorders responses to the problem.
Response: Added a short sentence about avoidance-coping in the text, to link it to theory for those who wants to read further about this after reading the article. Edits in line 35-39. “For example, people with poor self-esteem or specific self-esteem are found more likely to seek coping strategies who distract them from what is perceived as difficult, and to avoid uncomfortable stress and negative emotions by a shift in focus (Rosenberg et al., 1995; Silverstone & Salasi, 2003), often referred to as “avoldance-coping” in the literature (Taylor & Stanton, 2007).”
4. [Line 37] There are a lot of studies have found associations between certain coping strategies and levels of adjustment, reduced anxious and eating disorders symptomatology, and decreased emotional distress (Fullana. 2020; Mariani, et al., 2020; Violant-Holz et al., 2020). Coping strategies represents the cognitive and behavioral patterns to manage external and/or internal demands appraised as taxing or even exceeding the resources of individuals (Folkman & Lazarus, 1985).
Response: Some of the new evidence on adaptive coping behavior was further added to the text, to exemplify what adaptive coping behaviors may look like (line 45-48). “ More adaptive or effective coping-strategies in reducing emotional distress, and in turn leading to more positive mental health outcomes in the longer term is proposed to be adhering to a healthy diet, spending time pursuing hobbies or spending time outdoors (Fullana. 2020).»
5. As studies show the strong association between engaging in certain coping strategies and mental health problems during the pandemic (Guo et al., 2020; Sighn, 2020), I will recommend exploring coping styles within the population in that situation.
Response: Thanks for your great advice for enriching our study! Unfortunately the data was gathered at time before the COVID-pandemic took place, and the pandemic is therefore not relevant to mention in this study.
6. [Line 85] There are a lot of studies have found associations between certain coping strategies and maladaptive pattern of exercise. How can you specific that term “maladaptive” pattern of exercise. What will be exactly?
Response: to better describe the means of “maladaptive”, we replaced “maladaptive” with the term “dysfunctional” (Line 89).
7. [Line 107-108] I would recommend authors to mention those terms: “self-esteem” and “self-woth” the relation between them and the differences in order to identify the psychological eating disorders.
Response: This is described more in depth in section 1.2, and we decided not to add anything further about this in the section above that you referred to.
8. [Line 124-125] There are more recent studies about it from the last 3 years that I would recommend authors to mention such as (suggestions in the PDF).
Response: Thank you for giving advice regarding additional references, but we are not really sure if the recommended articles reflects the mentioned theme in line, since we focused on the behavioral outcomes related to self-esteem and stress, and not physiological aspects, like the suggested references indicate. Anyhow, we argue to keep the original references, mainly because they were carefully chosen, considering this are some of the most robust researchers and theorists on the field of self-esteem.
Materials and Methods:
1. Do the authors have a study protocol? The study protocol should be described in detail.
Response: All details concerning the method are reported in the manuscript.
2. It is necessary to include information about Design, Procedure and Sample (inclusion and exclusion criteria), Measuring Instruments, Ethical Considerations, You may have obtained an incomplete questionnaires, etc. ID number…..:2020). Interventionary studies involving animals or humans, and other studies require ethical approval must list the authority that provided approval and the corresponding ethical approval code. Please include the date and code register number of ethics committee.
Response: All details concerning the method are reported in the manuscript. No personal data whatsoever were collected; thus further formal approval was not required as per national regulations. The Norwegian Centre for Research Data, the National regulating body, state the following on their website: "If you are only going to collect anonymous data, then the project should not be notified to NSD." (https://eur01.safelinks.protection.outlook.com/?url=https%3A%2F%2Fwww.)
Are the Measuring Instruments adapted to the age and the gender population? Authors must justify their response. In this case more details are needed.
Response: all of the measure instruments have been validated for the respective ages and gender, and have been used on those age-groups in several published studies ()
4. The tables and figures are quite well clearly, and you indicate the analysis statistically significant.
Response: great!
Discussion
1. [Line 301-303] It is very interesting but, how these results can be globally interpreted?
Response: References to the two leading theoretical frameworks for understanding exercise dependence and the global aim of this study was added to outline how the findings could be more globally interpreted in terms of dysfunctional training behaviors and exercise dependence in the general population (line 302-307). “to further predict risk factors for developing an addiction to exercise. In line with our main hypothesis, and based on the theoretical frameworks for understanding Exercise Dependence by Freimuth et al., (2011) and Szabo & Egorov (2013), we found that higher levels of psychological distress and low levels of self-esteem were related to higher scores on measures of exercise dependence, as well as higher amounts of weekly exercise hours”
2. [Line 364-367] You could mention at least European cultural or some specific cultural that really affects it.
Response: To specify a cultural factor found to be affecting this we added a note on different cultural emphasis on appearance and body shape (Line 378-382). “Furthermore, cultural and national differences have also been reported to play a role in terms of prevalence of exercise dependence (Lindwall & Palmeira, 2009) as well as self-esteem and the cultural emphasis on appearance and body-shape (Cheng, 2000; Dotse & Asumeng, 2015), which must be taken into consideration when comparing our study to others.”
3. [Line 376-378] It is a major limitation because motivation is one of the tools that could help to study those variables. You could add some good references about it in order not to have such a major limitation.
Response: A few more findings from other research covering “our gaps” were added to the text, in order to provide an understanding of motivational aspects due to our limitations (changes in line 319-407). “A major limitation of our study was that exercise motivation included only these two categories: desire to improve performance in sports/races and health and fitness-related reasons, based on the findings of earlier studies showing that exercise for health reasons is found to be related to improvements in body image (Haugen, Ommundsen, & Seiler, 2013; Dotse & Asumeng, 2015; Smith-Jackson, Reel, & Thackeray, 2011) and self-esteem (Lindwall & Lundgren, 2005; Haugen, Ommundsen, & Seiler, 2013) during exercise interventions or regular physical activity. Other studies finds motives like exercising to lose weight, to improve one’s appearance, or to avoid the feeling of inferiority compared to others is related to reductions in self-esteem (Smith et al., 1998; Brattland-Sanda et al., 2011; Sibley et al., 2013). Hence, we should have included a third category addressing appearance-based exercise motivation to separate those exercising for health, and those exercising for appearance. The present results therefore need to be seen in light of studies reporting activity motivation (Sibley et al., 2013; De La Vega et al., 2016; Furnham et al., 2002) and context (Kleppang et al., 2018; Doré et al., 2018) as crucial for determining the psychological effects of that specific activity on self-esteem. Yet further research is needed, addressing the role exercise motivation plays for the relationship between exercise dependence, psychological distress, and self-esteem.”
4. In this part: limitations you can mention related with the type of methodology used. Authors must specify it.
Response: A few more reflections on limitations regarding to our research method was specified in the text (line: 416-419). “Further, our research was limited by the methodology of the present study itself, as cross-sectional questionnaires are seen as inappropriate when talking about causal relationships between a phenomena and related underlying mechanisms»
Conclusion
In my opinion, I think that Conclusion could give one or two suggestion to work it.
Response: In order to provide a clearer conclusion in line with our initial research question, we changed some aspects of the text. A third central finding in our study was further added to the conclusion in attempt to sum up our findings in a clearer manner (line 424-431). “First by expanding the excitant literature by showing a co-existence of low self-esteem and high levels of psychological distress in individuals with a high risk for developing exercise dependence. Second by showing a further co-existence between both low self-esteem, high levels of psychological distress and high risk for developing exercise dependence in individuals with weekly training amounts beyond 9 hours a week, even after adjusting for gender and age. Third, by supporting recent studies demonstrating an inverted U- relationship between weekly training amounts and mental health.»